# Decursinol Angelate Inhibits Glutamate Dehydrogenase 1 Activity and Induces Intrinsic Apoptosis in MDR-CRC Cells

**DOI:** 10.3390/cancers15143541

**Published:** 2023-07-08

**Authors:** Sukkum Ngullie Chang, Sun Chul Kang

**Affiliations:** Department of Biotechnology, Daegu University, Gyeongsan 38453, Republic of Korea; sukkumchang@gmail.com

**Keywords:** multidrug-resistant colorectal cancer cells (MDR-CRC), decursinol angelate (DA), glutamate dehydrogenase 1 (GDH1), apoptosis

## Abstract

**Simple Summary:**

Upregulated glutaminolysis is a typical hallmark of malignant tumors across different cancers. In our previous published paper, we proved the ability of decursinol angelate (DA) to bind to glutamate dehydrogenase 1 (GDH1) in the abortive complex of the ADP activation site and inhibit the activity of GDH. We also elucidated the significance of upregulated GDH in colorectal cancer (CRC) patients with a comparison of different subtypes of CRC and the survival probability for CRC across different races. As per the data collected by the Clinical Proteomic Tumor Analysis Consortium (CPTAC), we found that African Americans were highly susceptible to early death for CRC patients that had upregulated GDH. As specified in reports, glutamate dehydrogenase is a novel prognostic marker for predicting metastasis in colorectal cancer patients. The expression of glutaminolysis is highly upregulated in many different types of cancers such as lymph, cervix, lungs, breast and brain. In our present study, we evaluated the role of DA in inducing apoptotic cell death in multidrug-resistant colorectal cancer cells. Inhibiting glutaminolysis can be a good therapeutic approach for developing anti-cancer drugs.

**Abstract:**

Colorectal cancer (CRC) was the second most commonly diagnosed cancer worldwide and the second most common cause of cancer-related deaths in Europe in 2020. After CRC patients’ recovery, in many cases a patient’s tumor returns and develops chemoresistance, which has remained a major challenge worldwide. We previously published our novel findings on the role of DA in inhibiting the activity of GDH1 using in silico and enzymatic assays. No studies have been conducted so far to explain the inhibitory role of DA against glutamate dehydrogenase in MDR-CRC cells. We developed a multidrug-resistant colorectal cancer cell line, HCT-116^MDR^, after treatment with cisplatin and 5-fluorouracil. We confirmed the MDR phenotype by evaluating the expression of MDR1, ABCB5, extracellular vesicles, polyploidy, DNA damage response markers and GDH1 in comparison with parental HCT-116^WT^ (HCT-116 wild type). Following confirmation, we determined the IC_50_ and performed clonogenic assay for the efficacy of decursinol angelate (DA) against HCT-116^MDR^ (HCT-116 multidrug resistant). Subsequently, we evaluated the novel interactions of DA with GDH1 and the expression of important markers regulating redox homeostasis and cell death. DA treatment markedly downregulated the expression of GDH1 at 50 and 75 μM after 36 h, which directly correlated with reduced expression of the Krebs cycle metabolites α-ketoglutarate and fumarate. We also observed a systematic dose-dependent downregulation of MDR1, ABCB5, TERT, ERCC1 and γH2AX. Similarly, the expression of important antioxidant markers was also downregulated. The markers for intrinsic apoptosis were notably upregulated in a dose-dependent manner. The results were further validated by flow cytometry and TUNEL assay. Additionally, GDH1 knockdown on both HCT-116^WT^ and HCT-116^MDR^ corresponded to a decreased expression of γH2AX, catalase, SOD1 and Gpx-1, and an eventual increase in apoptosis markers. In conclusion, inhibition of GDH1 increased ROS production, decreased cell proliferation and increased cell death.

## 1. Introduction

Cancer is a genetic disease in which mutated cells of the body proliferate in an uncontrolled manner and aggressively spread to other parts of the body resulting in death. The causes of cancer are not understood completely; however, various factors are known to cause it (e.g., addiction to cancer-causing agents, obesity, genetic mutations) [1]. When such mutations and growth occur in the colon (large intestine) or rectum, it is known as colorectal cancer (CRC). CRC was the second most commonly diagnosed cancer worldwide and the second leading cause of cancer-related deaths in Europe in 2020 [2]. According to the National Center for Health Statistics, updated by the American Cancer Society, around 147,950 new cases of CRC were diagnosed in people and 53,200 individuals died from the disease in 2020. Approximately 4.4% of men (1 in 23) and 4.1% of women (1 in 25) will be diagnosed with CRC in their lifetime [3]. The risk of being diagnosed with CRC in a lifetime is similar in both men and women; males have a higher incidence rate, but women have a longer life expectancy. Additionally, age, race and habits also have a large influence on the risk. As per the data collected by the Clinical Proteomic Tumor Analysis Consortium (CPTAC) (http://ualcan.path.uab.edu, accessed on 3 June 2021) for GDH1 expression in CRC patients, apropos of different parameters such as different stages of cancer, stage 3 CRC patients were found to have higher GDH1 expression. Similarly, GDH1 expression in CRC patients according to the different races revealed African-American CRC patients to have higher GDH1 expression than the other races of CRC patients of Asian and Caucasian origin. African-American CRC patients were also found to be highly susceptible with a high mortality rate [4].

Treatments for CRC have rapidly advanced over the past decades. Nonetheless, the outcome of the treatment varies according to the molecular features of the tumor, location of the tumor, patient’s characteristics and so forth [5,6,7]. Some of the treatments include tumor resection surgery and adjuvant chemotherapy such as 5-fluorouracil (5-FU), cisplatin and oxaliplatin. These procedures for treating cancer have proven to be effective for patients suffering from CRC. Cisplatin and 5-FU have shown significant efficacy against rapidly proliferating tumor cells. However, in many cases, patients experience relapse, tumor recurrence and develop chemoresistance, which remain major challenges [8]. Chemotherapy acts as a selective pressure for aggressive cancer phenotypes which are less sensitive to drugs and often tend to develop mutations during exposure to drugs and thus evade drug-induced cell death [9]. Numerous studies have demonstrated that cancer cells develop resistance to cisplatin and 5-FU through various pathways such as a decrease in the cisplatin and 5-FU transporter proteins, activation of DNA repair pathways, enhanced inactivation by glutathione or metallothionein-mediated sequestration, modulation of DNA damage recognition, damage response and apoptosis evasion [10,11].

Altered cellular metabolism favoring proliferation, growth, maintenance and survival, irrespective of the type of cells or tissues involved, is observed across different cancers [12]. Cancer cells have an abnormally increased utilization of glucose uptake metabolism, a fundamental macronutrient through which energy is harnessed as ATP by the oxidation of carbon bonds and lactate production, a metabolic process vitally essential for mammalian life [13,14]. However, it is abnormally upregulated in cancer cells even in the presence of oxygen, which is known as the Warburg effect. Additionally, cancer cells also have another upregulated metabolic pathway to meet the requirements of proliferating cells. Accumulating evidence has shown that cancer cell proliferation and tumor growth have a direct correlation with elevated glutamine metabolism, known as glutaminolysis [15]. Glutamine is the most abundant non-essential amino acid. Glutamine in cancer cells is catabolized to glutamate with the help of the enzyme glutaminase (GLS/GLS2). Glutamate is converted into α-ketoglutarate via glutamate dehydrogenase (GDH) and the co-enzyme NADP^+^, producing a Krebs cycle intermediate through the activity of different enzymes [16].

In this study, we used a pyranocoumarin compound, decursinol angelate (DA), found in different plants, such as *Seseli grandivittcitum*, *Angelica gigas Nakai* and *Angelica acutiloba kitagawa*, belonging to the family Umbellifereae [17]. Numerous studies have reported the anti-inflammatory, anti-angiogenic, anti-amnesic and anti-cancer activity of decursinol angelate (DA) [18]. We previously evaluated the role of DA in inducing apoptotic cell death in melanoma in vitro and in vivo [19]. We also published a novel finding for the role of DA in inhibiting the activity of GDH1 [4]. However, no studies have been conducted till now to show the inhibitory role of DA against glutamate dehydrogenase. We developed a multidrug-resistant HCT-116^MDR^ cell line using the parental HCT-116 by doubling the dose concentration of cisplatin and 5-fluorouracil over a period of three months, until the cells had become completely resistant at a specific dose range that would normally have caused massive cell death. Our study elucidated the role of DA in triggering cell death in multidrug-resistant human CRC with particular emphasis on glutamate dehydrogenase.

## 2. Materials and Methods

### 2.1. Chemicals, Antibodies and Reagents

Decursinol angelate (DA) (PubChem CID: 776123) was purchased from Cayman Chemicals (Item no: 25212); RPMI-1640, 3-(4,5-dimethylthiazol-2-yl)-2,5-diphenyltetrazolium bromide (MTT), dimethyl sulfoxide (DMSO), ethidium bromide and mitotracker were purchased from Sigma (Sigma-Aldrich, St. Louis, MO, USA); and glutamate dehydrogenase (EC number: EC 1.4.1.2.). Fetal bovine serum was purchased from Gibco, USA. The complete details of all the primary and secondary antibodies used in this experiment can be found in a Appendix A. All the other solvents used for the study were of highest grade and supplied by Sigma (Sigma-Aldrich, St. Louis, MO, USA).

### 2.2. Establishment of MultiDrug-Resistant HCT-116 Colorectal Cancer Cell Lines

Parental HCT-116^WT^ cells were purchased from ATCC (Rockville, MD, USA) and maintained at 37 °C (5% CO_2_) in RPMI 1640 medium (Gibco BRL, Gaitherburg, MD, USA) supplemented with 10% FBS and 1% penicillin–streptomycin (Gibco, USA). The process for selecting the cell line used and establishing the multidrug-resistant cell line was carried out in our departmental laboratory [20,21]. Briefly, the HCT-116 cell line was allowed to grow and reach 70% confluence in a 25 cm^2^ flask. Later, the cells were treated with 0.5 μM of cisplatin (cis) and 5-fluorouracil (5-fu) for 24 h, followed by three days of incubation in fresh medium, allowing the surviving cells to grow and adapt to an obvious colony. After the completion of three cycles of drug treatment, the doses were doubled and the procedure was repeated until the final level of resistance to the drugs was achieved. At the end of 90-day period, the cells had become resistant to 10 μM cis and 5-fu. All the cell lines were maintained as a monolayer in the complete medium. The selected colonies were amplified in the presence of all drugs until confluency before the drug dose was increased in multiples of two for the next round of selection. The multidrug-resistant (MDR) subline was maintained at 10 μM cis and 5-fu and was denoted as HCT-116^MDR^.

For experimental purposes, 70–80% confluent HCT-116^WT^ and HCT-116^MDR^ cells were extracted using RIPA buffer and centrifuged at 12,000 rpm for 15 min and the supernatant was collected for analysis.

### 2.3. Cell Culture and In Vitro Assays

HCT-116^WT^ (passage number 2, ATCC) and HCT-116^MDR^ cells were cultured in RPMI-1640 supplemented with 10% fetal bovine serum (Gibco, USA) and 1% penicillin–streptomycin (Gibco, USA). Cells were carefully maintained in a CO_2_ incubator (5% CO_2_) at 37 °C and constantly checked. For performing the experiment, cells were cultured in a 60 × 15 mm cell culture petri dish at a density of 1 × 10^5^ cells/well. When the cells had reached 60–75% confluence, the cells in different petri dishes were grouped according to the treatment and non-treatment plates, and also the different concentrations of the cells were treated with 50 and 75 μM DA for both parental HCT1-116^WT^ and HCT-116^MDR^. After 36 h, the cells were used for staining and evaluation of different markers.

### 2.4. MTT Assay and Morphological Assessment

For performing the cell viability assay, we used an MTT assay and determined the percentage of live and dead cells after treatment with DA. We referred to the previously published paper [22]. HCT-116^WT^ and HCT-116^MDR^ cells were seeded in a 96-well flat-bottom microtiter plate at a density of 1 × 10^4^ cells per well, mixed with 200 μL of RPMI-1640 media and allowed to grow and become confluent for 24 h at 37 °C in a CO_2_ incubator. Once the HCT-116^MDR^ cells had reached 70–80% confluence, the RPMI media was replaced with fresh medium and the cells were treated with different concentrations of DA (25, 50, 75, 100 μM) and incubated for 36 h at 37 °C. Subsequently, after the completion of the drug treatment period, the culture medium was replaced with 100 μL of fresh RPMI media along with 10 μL of MTT working solution (5 mg/mL in PBS) added into each of the wells. Next, we incubated the plates for 4 h at 37 °C. Later on, the medium was aspirated, and the formazan crystals were solubilized with the addition of 50 μL DMSO per well and allowed to incubate for 30 min. Lastly, formazan crystals (purple color) that were fully dissolved in DMSO were quantified at a wavelength of 540 nm using an ELISA plate reader. The percentage of live and dead cells was calculated using the following formula:Cell viability (%) = (Treated group/control group) × 100%

### 2.5. Lactate Dehydrogenase Assay

HCT-116^WT^ and HCT-116^MDR^ cells were treated with different concentrations of DA (25, 50, 75, 100 μM) and incubated for 36 h for induction of cellular cytotoxicity. After the treatment period, an LDH assay kit (Sigma-Aldrich, St. Louis, MO, USA) was used to determine the cellular cytotoxicity and the experiment was carried out according to the manufacturer’s instructions.

### 2.6. ATPase Activity Assay

For evaluation of an ATPase activity assay on the HCT-116^MDR^ cells, the experiment was carried out according to the manufacturer’s instructions (ATPase activity assay Kit, Biovision). The details of the assay kit used can be found in Appendix A.

### 2.7. Intracellular Metabolite Measurements

For measuring the intracellular levels of alpha-ketoglutarate (αKG) and fumarate, 2.3 × 10^6^ HCT-116^MDR^ cells after 36 h treatment with DA or cis+ 5-fu were homogenized in PBS. The supernatants from the different groups were collected and the proteins were finally extracted by using the 10 KD Amicon Ultra Centrifugal Filters (Millipore). Finally, we followed the protocol and procedure mentioned in the commercially available kit (Biovision). The details of the assay kits used can be found in Appendix A.

### 2.8. Clonogenic Assay

For evaluating the cell proliferation and colony forming assay, we performed the experiment as per the protocol mentioned in [23]. Briefly, HCT-116^WT^ and HCT-116^MDR^ cells were seeded in RPMI-1640 medium using 12-well tissue culture plates (BD Falcon, CA, USA) at a seeding density of 3 × 10^3^ cells/well. After the cells had attached to the plates, HCT-116^WT^ and HCT-116^MDR^ cells were exposed to different concentrations of DA (50 and 75 μM) or cis+ 5-fu (10 μM) for 36 h. After the end of the incubation period, the cells were fixed in 6% (*w*/*v*) glutaraldehyde, and later stained with 0.1% (*w*/*v*) crystal violet solution, and finally images were taken.

### 2.9. Determination of Morphological Changes and Apoptosis by Fluorescence Staining

HCT-116^WT^ and HCT-116^MDR^ cells were cultured in RPMI-1640 medium and were treated with varying concentrations of DA (50 and 75 μM) or cis+ 5-fu (10 μM) for 36 h. Following treatment, the cells were washed with 1 mL of PBS twice and allowed to fix in a cold 4% formaldehyde solution for 10 min. After fixing, cells were washed with PBS, and Hoechst 33,342 (1 mg/mL) was added onto the plates and incubated at 37 °C for 10 min in the dark for evaluation of DNA damage after DA treatment [24,25]. The cells were washed again with PBS, and the intensity of the resulting fluorescence was detected using an Olympus BX50 fluorescence microscope.

### 2.10. Immunofluorescence Staining

For performing immunofluorescence staining, HCT-116^WT^ and HCT-116^MDR^ cells were seeded in glass cover slips at a density of 1 × 10^5^ per well and were allowed to attach and become confluent [26,27]. When the cells had become confluent, we treated the cells with varying concentrations of DA (50 and 75 μM) or cis+ 5-fu (10 μM) for 36 h, and they were fixed using 4% paraformaldehyde for 10 min at RT. Following fixation, cells were washed with ice-cold PBS thrice (for 3 min each time). Next, we performed the antigen retrieval step and heated the slides with the cells by heating at 95 °C for 10 min immersed in antigen retrieval buffer. Next, all the slides were washed in PBS and incubated with 0.1% triton X-100 (10 min) for making the cells permeable for antibody interaction after treatment. Later, the slides were immersed in blocking buffer (1% BSA in PBST) for 30 min. After the completion of the incubation period, the slides were further treated with the primary antibody of interest (GDH1, ERCC1, Gpx-1) and incubated overnight at 4 °C. After overnight incubation, the slides were washed in PBS three times (for 3 min each time) and further incubated with FITC-secondary antibody for 1 h at RT, which was followed by counterstaining the cells with DAPI (1 μg/mL) for 5 min. Next, coverslips were added onto the slides and were sealed using a mounting medium and observed under an Olympus BX50 fluorescence microscope (600X).

### 2.11. Flow Cytometry Analysis

For analyzing the occurrence of cellular apoptosis through flow cytometry, we treated the HCT-116^MDR^ cells with different concentrations of DA (50 and 75 μM) or cis+ 5-fu (10 μM) for 36 h [28]. After the incubation period was completed, the cells were extracted using trypsin and mixed with PBS, and finally we determined the occurrence of different stages of apoptosis using an Annexin V/PI apoptosis detection kit as per the manufacturer’s protocol.

### 2.12. Isolation of Proteins

For extracting the cytosolic protein, the cells were trypsinized and lysed by adding RIPA buffer (1:100 dilutions of protease and phosphatase inhibitor) followed by vigorous shaking and incubating the cell tubes on ice for 10 min. Next, the cells were centrifuged at 12,000 rpm and 4 °C for 15 min. After the completion of centrifuging, supernatant was collected in a different centrifuge tube and labelled as per the groupings, and it was stored at −70 °C until further analysis. The concentration of the total present in the supernatant was quantified by using a Bradford assay.

### 2.13. Western Blotting

For performing Western blotting analysis, the supernatants were quantified for total protein and loaded onto an SDS-polyacrylamide gel electrophoresis unit and the proteins were allowed to resolve by electrophoresis at 100 V. After the proteins had separated on the gels, the gels were transferred to polyvinylidene difluoride membranes (PVDF) [29]. Next, when the proteins had transferred to the PVDF membranes, the membranes were incubated in 3% BSA in TBST solution for 1 h for blocking. After completion of the blocking, the membranes were probed with the specific primary antibodies of interest and incubated overnight at 4 °C. Following the completion of incubation time, the blots were further washed three times with wash buffer TBST (for 5 min each time) and probed with a secondary antibody for 1 h at RT. After the completion of secondary antibody incubation, the blots were washed again three times. The PVDF membrane blots having proteins were further visualized through an enhanced chemiluminescence Western blotting detection reagent (Amersham Biosciences Inc., Piscataway, NJ, USA).

### 2.14. siRNA Transfection

HCT-116 and HCT-116^MDR^ cells were seeded in a 6-well plate and transfected with 25 pM siRNA against GDH1 (integrated DNA technologies) bearing the sequence (5′-rCrUrUrArCrUrArUrAr ArUrU rArUrG rArUrA, 5′-rUrGrG rArCrU rGrUrA rUrCrA rArArA rUrUrA). The cells were treated with DA (75 μM) for one group after siRNA knockdown, and after the completion of the necessary treatment and incubation period, we followed the protein extraction protocol mentioned in the previous sections and performed Western blotting analysis.

### 2.15. Statistical Analysis

For evaluating and determining the quantitative results and the significance of the studied results, the values were expressed as a mean standard deviation (SD) of experiments performed three times. The statistical significance and the differences in the experimental groups of WT ^Cis+5fu^ and HCT-116^MDR^, MDR^DA50^, MDR^DA75^, NAC groups and siRNA groups were calculated by using one-way analysis of variance (ANOVA) with Tukey’s comparing all pair of columns and Student’s *t*-test from Prism software (* represents *p*-values < 0.05, ** represents *p*-values < 0.01 and *** represents *p*-values < 0.001).

## 3. Results 

### 3.1. Phenotypic and Molecular Differences between Parental HCT-116^WT^ and HCT-116^MDR^ Cells

In our study, HCT-116^MDR^ cells in comparison with the parental (wild type) HCT-116^WT^ had higher expression and enrichment of lipid ceramide in extracellular vesicles (EVs) (Figure 1A), a direct indication of the role of EVs in conferring multidrug-resistant (MDR) phenotype. Nuclear staining analysis of HCT-116^WT^ and HCT-116^MDR^ also revealed higher expression of polyploidy giant cells or multinucleated cells in MDR cells compared to HCT-116^WT^ cells (Figure 1B,C). This may be an indication that MDR cells had multiple nuclei accumulated in a single cell, which is another hallmark of cells becoming resistant to certain drugs [30]. Similarly, we observed that the expression of MDR1 (*p* < 0.0002) and ABCB5 (*p* < 0.0001) were upregulated in MDR cells compared to HCT-116^WT^ parental cells. The markers for cell proliferation, such as p-ERK 1/2 (*p* < 0.0057), were more numerous in MDR cells. There was no observable difference in Ki-67 expression (Figure 1D). The markers for DNA damage response, such as γ-H2AX (*p* < 0.0020), were also upregulated in MDR cells in comparison with HCT-116^WT^ cells. ERCC1 expression (*p* < ns) was upregulated in both HCT-116^WT^ and HCT-116^MDR^ cells (Figure 1E).

### 3.2. The Expression of Glutamate Dehydrogenase 1 in Parental HCT-116^WT^ and HCT-116^MDR^ CRC Cells

In order to evaluate the expression of glutamate dehydrogenase, a mitochondrial enzyme involved in glutamine metabolism, we performed immunofluorescence staining of GDH1 on HCT-116^WT^ and HCT-116^MDR^ CRC cells. The fluorescence intensity of GDH1 was slightly higher in parental HCT-116^WT^ cells in comparison with HCT-116^MDR^ cells. However, both types of human CRC cells highly expressed the enzyme involved in glutaminolysis (Figure 2A). Concomitantly, the expression of GDH1 was directly correlated with the expression of an ROS scavenging enzyme, glutathione peroxidase, as shown in Figure 2B. We also compared the expression of ATPase activity and Krebs cycle metabolites. The concentrations of α-ketoglutarate (*p* < 0.0003) and fumarate (*p* < 0.0612) for control were 102.5 μM and 41.5 μM, whereas MDR cells had concentrations of 96 μM and 39.5 μM (Figure 2C–E).

### 3.3. The Effect of Decursinol Angelate-Induced Cytotoxicity in HCT-116^MDR^ CRC Cells

To evaluate the effect of DA on MDR cells, HCT-116^MDR^ cells were treated with varying concentrations of DA (25, 50, 75 and 100 μM) and we evaluated the percentage of cell viability and lactate dehydrogenase (LDH) release at different time points of 24 h, 36 h, 48 h and 72 h. As shown in Figure 3A, the cells treated with either combination of cis and 5-fu (10 μM) or DA (50 and 75 μM) showed more cell shrinkage, fragmented nucleus and membrane-bound apoptotic bodies, and overall phagocytosis of the neighboring cells. Next, we performed live and dead cell estimation and a membrane damage assay after the treatment with DA at different time periods. The cell viability at 24 h for DA 50 μM and 75 μM was 83.2% and 74.16%, respectively, whereas the LDH release at 24 h for the same dose was 18.11% and 25.61%. Cell viability at 36 h was 70.91% and 50%, respectively, and the LDH release was 25.66% and 31.54% (Figure 3B,C). Similarly, we saw a systematic downregulation of cell death and LDH release at both 48 h and 72 h in a dose-dependent and time-dependent manner. As shown in Figure 3D, we observed that HCT-116^WT^ and HCT-116^MDR^ cells had high proliferation and colony-forming rates, which drastically minimized at 36 h incubation with DA. It was observed that most of the MDR cells had detached from the cell culture plates and were undergoing cell death, as shown in the morphological images in Figure 3A.

### 3.4. DA Downregulated MDR Phenotype, DNA Damage Repair, and Inhibited GDH1 Activity

In our study, we used DA, a small coumarin compound extracted from plants, which might hold the potential to inhibit the development of drug resistance or promote cell death in MDR CRC cells that had evaded apoptosis. After treatment with DA or Cis+ 5-fu for 36 h, we performed Western blotting to evaluate the markers for MDR phenotype such as MDR1 and ABCB5. We observed a systematic dose-dependent downregulation of MDR1 (*p* < 0.0001), ABCB5 (*p* < 0.001) and TERT (*p* < 0.001) (Figure 4B). DNA damage repair markers were upregulated in MDR groups. However, DA treatment significantly downregulated the expression of ERCC1 (*p* < 0.0001) and γH2AX (*p* < 0.0002) (Figure 5A,B). Figure 4A showed DA treatment significantly downregulated the expression of GDH1. Western blotting analysis also revealed the dose-dependent downregulation of GDH1 in DA-treated groups (*p* < 0.0001). We also noticed a corresponding downregulation in the activity of ATPase (*p* < 0.0018) (Figure 4C), α-ketoglutarate (*p* < 0.0001) and fumarate (*p* < 0.0001) (Figure 4D,E).

### 3.5. DA Destabilized Redox Homeostasis Regulated by GDH1 in HCT-116^MDR^ CRC Cells

The results after DA treatment showed the expression of lower levels of antioxidants such as SOD1 (*p* < 0.0001), SOD2 (*p* < 0.0001), catalase (*p* < 0.0001), Gpx-1 (*p* < 0.0001), GR (*p* < 0.0001), GST (*p* < 0.0001), HO-1 (*p* < 0.0001) and NRF2 (*p* < 0.0001) (Figure 6A, B), and maneuvering the cells towards DA induced cell death.

### 3.6. DA Triggered Intrinsic Apoptosis in HCT-116^MDR^ CRC Cells

After treatment with DA for 36 h, we noticed a sharp rise in the markers mediating intrinsic apoptosis similar to the cisplatin and 5-fluorouracil treatment groups of HCT-116^WT^ cells (Figure 7B). Additionally, TUNEL assay also provided insights about the cells undergoing apoptosis. The typical characteristics of apoptosis such as cell shrinkage, loss of DNA and degradation of cytosolic proteins were clearly observed in the DA treatment groups of MDR^50^ and MDR^75^ (Figure 7A). Furthermore, we performed flow cytometry to confirm the activation of apoptosis by DA in MDR cells (Figure 7C). The percentages of live cells were calculated from Q4 in the dot blots depicted in Figure 7C. MDR (untreated cells) had a live cell percentage of 95.3%, whereas the DA treatment groups of MDR^50^ and MDR^75^ had live cell percentages of 82.4 and 78.13, respectively (Figure 7D). Similarly, the data for the apoptotic events can be found in (Figure 7E–G). These results confirmed that DA triggered intrinsic apoptosis in MDR CRC cells.

### 3.7. NAC Attenuated DA-Induced Alteration of Antioxidant, DNA Repair and Apoptosis Pathways in HCT-116^MDR^ CRC Cells

Studies have reported NAC to possess antioxidant and anti-cancer properties in different experimental disease models. Invariably, we noticed similar molecular restorations of homeostasis after treatment with NAC (10 mM) for 24 h in MDR CRC cells. The representative blots (Appendix A) clearly showed that NAC had reparative effects on the MDR cells. The concentrations of DNA damage repair markers such as ERCC1 (*p* < 0.0001) and γH2AX (*p* < 0.0001) were higher in the NAC group compared with the other groups. Antioxidant enzymes such as catalase (*p* < 0.0041) and gpx-1 (*p* < 0.0004) were also higher in the NAC group compared with the co-treatment group of NAC and NAC+ DA (75 μM). The concentration of glutaminolysis enzyme GDH1 was also restored in the NAC group. Intrinsic apoptosis markers were downregulated in the NAC group (Appendix A). Our results were further substantiated through the flow cytometry analysis for apoptosis which also revealed similar findings (Appendix A), in contrast to MDR + DA (75 μM) (Figure 7C–G) which had higher apoptotic occurrence. In conclusion, NAC alleviated the damage in MDR cells induced by treatment with DA.

### 3.8. Silencing GDH1 Impaired Metabolic Activity and Cell Proliferation and Increased Cell Death in HCT-116^WT^ and HCT-116^MDR^ CRC Cells

We wanted to evaluate the role of glutaminolysis and glutamine metabolism in human CRC. To better understand its importance in cancer, we used the parental wild-type (HCT-116^WT^) and the multidrug-resistant (HCT-116^MDR^) human CRC cell lines. We blocked the enzyme involved in the conversion of glutamate to α-ketoglutarate by using siRNA for GDH1. After knockdown of the GDH1 enzyme from both HCT-116^WT^ and HCT-116^MDR^, we performed a colony-forming assay and evaluated the extent of cell proliferation. GDH1 siRNA was also able to inhibit the cell proliferation and colony forming, although co-treatment of GDH1 siRNA + DA (75 μM) was more efficient (Figure 8A,E). Expression of intracellular ATP was also downregulated in the GDH1 siRNA groups of HCT-116^WT^ and HCT-116^MDR^ in comparison with the control (mock) groups (Figure 8B,F). Additionally, Krebs cycle intermediates such as intracellular fumarate and α-ketoglutarate were also systematically downregulated in the GDH1 knockdown groups of both HCT-116^WT^ and HCT-116^MDR^ (Figure 8C,D,G,H). We also quantified the different protein expressions (Figure 8I,J). Additionally, flow cytometry analysis for the HCT-116^MDR^ GDH1 siRNA knockdown showed that silencing GDH1 after 36 h resulted in an increase in apoptotic events (Figure 8K–O). From the above obtained results, we concluded that defects in the metabolic activity by inhibition of GDH1 were involved in increased ROS production, a decrease in cell proliferation and increased cell death.

## 4. Discussion

Drug-resistant cancer phenotypes have been a major obstruction that has limited the efficacy of chemotherapeutic drugs in CRC treatment [31]. The puzzle that lies before us has been dauntingly insurmountable, and cancer as a whole has continued to be the leading cause of death worldwide. Many studies have replicated the development of drug-resistant cancer cell models and these have helped us to understand the molecular mechanisms involved in the development of drug resistance [32]. However, clinical trials to overcome drug resistance have not been able to find a novel strategy to prevent it or to extend the life of many cancer patients. Cancer cells’ ability to become resistant to cytotoxic drugs was first reported by Dano in 1973 [33]. Subsequent research work has also revealed that the reduced drug permeation in multidrug-resistant cells was directly associated with the presence of a cell surface receptor known as multidrug resistance protein 1 (MDR1) [34]. In that study, MDR1 was found to be an ATP-binding cassette (ABC) transporter protein which functioned as an efflux pump of the drugs [35]. ATPase-binding cassette sub-family B member (ABCB5), another novel ABC multidrug-resistant transporter protein, was also found to arbitrate the fusion of cells, the functioning of stem cells and also vasculogenic plasticity [36]. MDR1 and ABCB5 stood out among other ABC transporter proteins by conferring the strongest resistance to a vast array of compounds and chemotherapeutic drugs. Hence, elevated expression of MDR1 and ABCB5 confirms a multidrug-resistant phenotype in almost all cancer cells. Cancer cells develop resistance to chemotherapeutic drugs (cisplatin, 5-fluorouracil, etc.) due to a myriad of alterations in the characteristics and expression of genes and proteins. Some of these alterations include cross-resistance to many structurally similar drugs, decreased accumulation of cisplatin and 5-fu due to reduced expression in their transporter proteins, and overexpression of DNA damage repairs, cell cycle, molecular chaperones, transcription factors, transporters, protein trafficking, oncogenes, GSH, cytoskeletal proteins, mitochondria, resistance to apoptosis, etc. [37,38]. According to the National Institutes of Health statistics on cancer, more than 1.2 million people develop cancer every year, and every 30 s a new cancer is diagnosed in the United States alone. Our goal should be to identify any natural compounds possessing potent anticancer properties that can contribute to the fight against cancer. The discovery of artemisinin, a sesquiterpene lactone isolated from sweet wormwood, Artemisia annua, saved millions of lives across the globe and gave hope to millions of people suffering from malaria. Similarly, recent studies have demonstrated that numerous small-molecule inhibitors, such as curcuminoids, have been identified as potential inhibitors of MDR1 for preventing the development of drug resistance [39,40]. In this search, recent studies have been conducted on small molecules that act as inhibitors or possess the ability to prevent the development of MDR [41]. 

In our previous study, we evaluated the cytotoxic role of DA in human CRC HCT-116 cells [19]. For this study, we developed a unique multidrug-resistant human CRC cell line from the parental HCT-116^WT^ cell line. The development of this multidrug-resistant cell line for cisplatin and 5-fluorouracil was discussed in the Materials and Methods section (Section 2.2). After the cells were fully resistant to cis and 5-Fu (10 μM), to evaluate cell damage we used cell viability and LDH release, a widely used technique for evaluating the efficacy of therapeutic compounds as cell viability predicts the number of live and dead cells and LDH is an enzyme that is released when the cell membrane is damaged [42]. Concomitantly, the hallmark characteristics of cells developing resistance to cisplatin and 5-fluorouracil were followed by increased expression of excision repair cross-complementation group 1 (ERCC1) and gamma-H2A histone family member X (γH2AX) and increased ATPase activity [43]. We also evaluated the important antioxidant markers. As per reports, in any type of cancerous or non-cancerous cells, a balanced system has been evolved over a period of many cellular modifications that helps the cells to neutralize the extra ROS through the antioxidant system of enzymes such as glutathione peroxidases, superoxide dismutase and catalase, as well as the non-enzymatic antioxidants which collectively reduce the oxidative state [44]. We observed a dose-dependent downregulation of the marker for MDR after treatment with DA in a dose-dependent manner. Another important marker that is highly upregulated is glutamine. It is one of the most important amino acids in the body and has multiple distinct functions in promoting cancer cell proliferation, tumor growth by generating energy, and biosynthesis of important biomolecules such as amino acids, fatty acids, purine and pyrimidines, and also in controlling the redox homeostasis [45]. Glutamate dehydrogenase (GDH) is a mitochondrial hexameric enzyme and is found in all organisms involved in the conversion of glutamine. Its function is to catalyze the reversible oxidative deamination of glutamate to α-ketoglutarate and ammonia with the help of the coenzyme NAD(P)^+^ to NAD(P)H (Graphical abstract). Through this conversion, it helps to fuel the cancer cells by contributing to the tricarboxylic acid cycle (TCA cycle) by refilling the enzyme catalytic intermediates involved in the energy production cycle carrying acetyl-CoA. Some of the key anaplerotic substrates are glutamine/glutamate, pyruvate, precursors of propionyl-CoA, amino acids and C95 ketone bodies [46]. Studies have reported that GDH1 modulates redox homeostasis in cancer cells by converting glutamate to α-ketoglutarate, a metabolite of the Krebs cycle, which in turn is converted to fumarate that binds directly to the mitochondrial glutathione peroxidase-1, a potent ROS scavenger that promotes redox homeostasis and tumor growth [47]. In our study, we were able to clearly observe that DA was directly involved in the downregulation of GDH1. Through our molecular dynamics simulation and enzymatic assay study, we also showed the binding of DA with GDH1, thus inhibiting the activity of GDH1 [4].

NAC has been found to possess anti-genotoxic and anti-carcinogenic properties in various experimental disease models [48,49]. Studies have shown that NAC is able to modulate the different stages of cancer. Prevention of cancer by administering NAC has been effective in various animal models [50]. Likewise, oral administration of NAC prevented damage and alterations in the DNA, DNA–protein cross-linking and nucleotide modifications in a rat lung model study [51]. Furthermore, NAC has prevented the formation of carcinogen–DNA adducts and inhibited the formation of tumors in an in vivo model [52]. NAC is also able to alleviate drug-induced apoptosis and cell death in vitro through its ROS scavenging property, mitigating the genes and proteins involved in the mitochondrial respiratory chain system, and so forth [53]. In our study, we observed that NAC was able to increase the protein expression of important antioxidants and reduce the expression of apoptosis execution markers. We also observed that DA triggered intrinsic mediated apoptosis in MDR cells that had evaded apoptosis by becoming resistant to cisplatin and 5-fluorouracil. Conclusively, glutaminolysis played a pivotal role in cancer cell metabolism, cell signaling and cell proliferation and thus it can be a good therapeutic target for many different types of cancers. 

## 5. Conclusions

This pilot study on a small coumarin compound, decursinol angelate, showed inhibition against glutamate dehydrogenase 1, which regulates the glutaminolysis involved in fueling cancer cells. Additionally, multidrug-resistant markers such as MDR1 and ABCB5 were downregulated after DA treatment in human MDR CRC cells. the results obtained thus far from this study proved DA to be a potent small-molecule natural compound possessing anti-cancer properties against both parental human CRC and multidrug-resistant CRC cells. Further studies need to be carried out on using DA as a GDH1 inhibitor possessing anticancer properties that can prevent the development of MDR phenotypes.

## Figures and Tables

**Figure 1 cancers-15-03541-f001:**
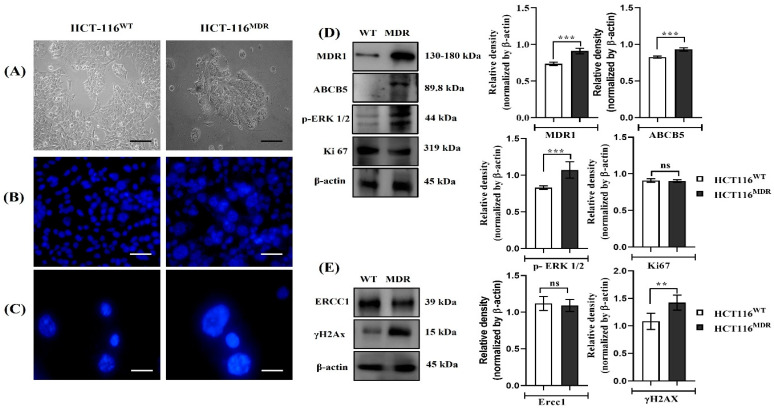
Evaluation and comparison of parental HCT-116^WT^ and HCT-116^MDR^ CRC cells. (**A**) Comparison of morphological characteristics of HCT-116^WT^ and HCT-116^MDR^; (**B**) Hoechst staining of HCT-116^WT^ and HCT-116^MDR^ (magnification: 200X); (**C**) Zoomed Hoechst staining of HCT-116^WT^ and HCT-116^MDR^ (magnification 1000×); (**D**) Western blotting analysis of MDR phenotype and cell proliferation pathway; (**E**) Western blotting analysis of DNA damage repair markers [scale bar = 100 µm]. Densitometry analysis for all the proteins was normalized with β-actin and measured using Image J software (version 1.51k). The representative data shown here are the means ± S.D. from three independent experiments where ** *p* < 0.01 and *** *p* < 0.001. HCT-116^WT^ vs. HCT-116^MDR^ calculated through ANOVA prism. Original western blots are presented in Appendix A.

**Figure 2 cancers-15-03541-f002:**
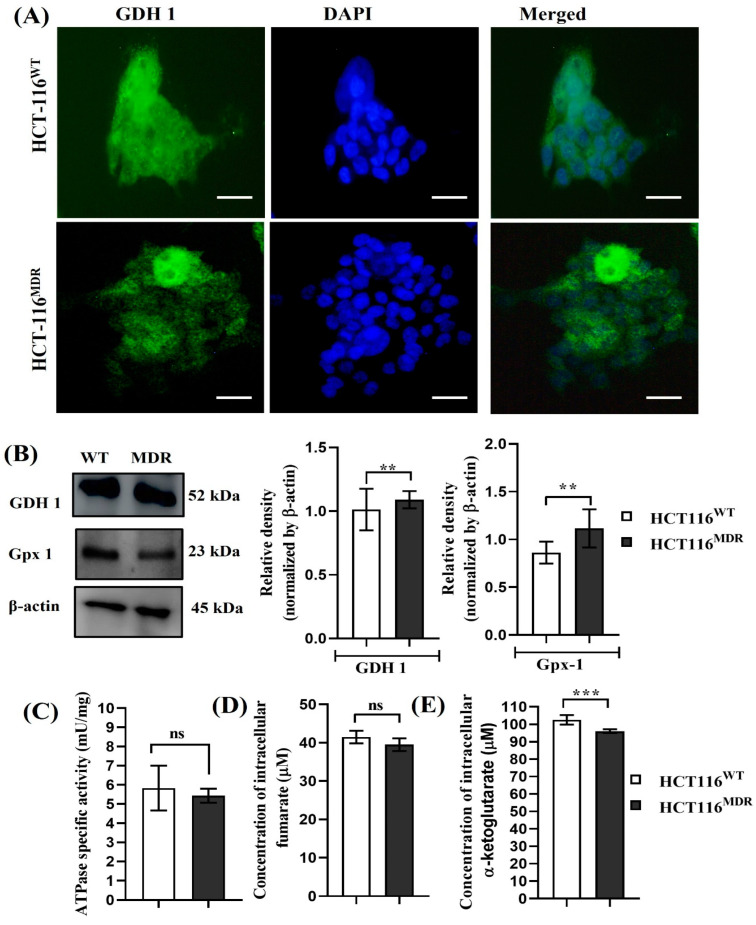
Expression of GDH1 and its metabolites in HCT-116^WT^ and HCT-116^MDR^ CRC cells. (**A**) Immunofluorescence staining of GDH1 on HCT-116^WT^ and HCT-116^MDR^ (magnification: 600×, scale bar: 25 μm); (**B**) Western blotting analysis of GDH1 and Gpx-1 in HCT-116^WT^ and HCT-116^MDR^; (**C**) ATPase activity was measured in HCT-116^WT^ and HCT-116^MDR^; (**D**) expression of intracellular fumarate levels in HCT-116^WT^ and HCT-116^MDR^; (**E**) expression of α-ketoglutarate levels in HCT-116^WT^ and HCT-116^MDR^. Densitometry analysis for all the proteins was normalized with β-actin and measured using Image J software. The representative data shown here are the means ± S.D. from three independent experiments where ** *p* < 0.01 and *** *p* < 0.001. HCT-116^WT^ vs. HCT-116^MDR^ calculated through ANOVA prism. Original western blots are presented in Appendix A.

**Figure 3 cancers-15-03541-f003:**
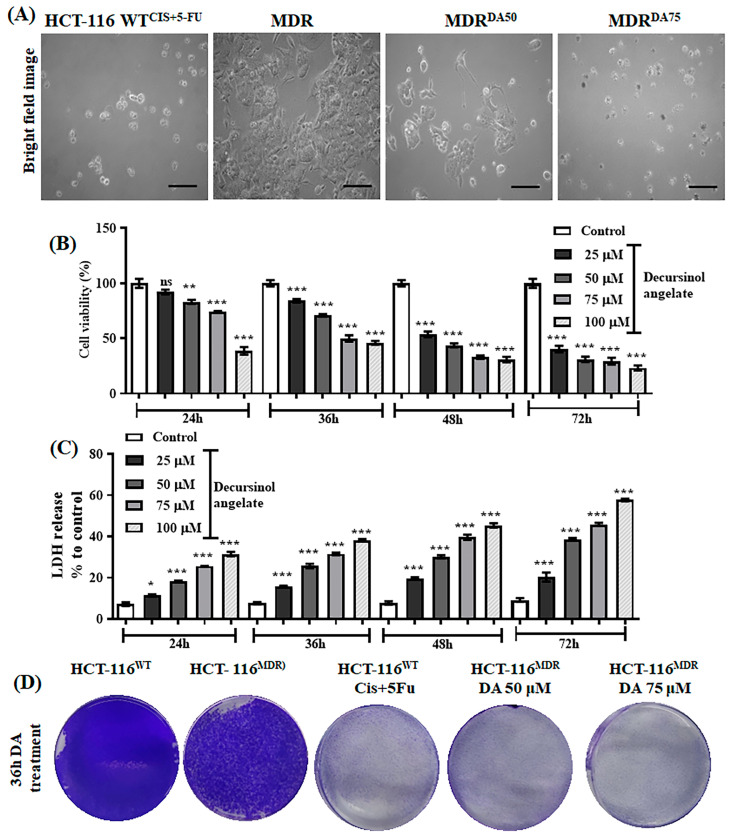
Time-dependent evaluation after DA treatment in HCT-116^MDR^ CRC cells. (**A**) Bright field images of HCT-116^MDR^ after 36 h treatment with DA (magnification: 200×, scale bar = 100 µm); (**B**) percentage of cell viability in HCT-116^MDR^ cells after treatment with different concentrations of DA at different time points; (**C**) percentage of LDH release in HCT-116^MDR^ cells after treatment with different concentrations of DA at different time points; (**D**) evaluation of colony-forming rate and cell proliferation in HCT-116^MDR^ cells after treatment with different concentrations of DA for 36 h. The representative data shown here are the means ± S.D. from three independent experiments where * *p* < 0.05, ** *p* < 0.01 and *** *p* < 0.001. HCT-116^MDR^ vs. all other groups calculated through ANOVA prism.

**Figure 4 cancers-15-03541-f004:**
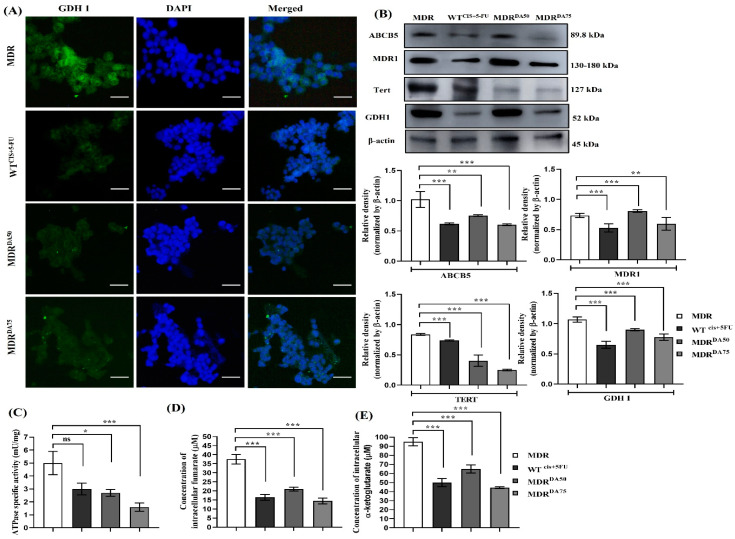
Effects of DA on the expression of MDR phenotype, GDH1 and its metabolites in HCT-116^MDR^ CRC cells. (**A**) Immunofluorescence staining of GDH1 in HCT-116^MDR^ cells after treatment with DA for 36 h (magnification: 600×, scale bar: 25 μm); (**B**) Western blotting analysis after treatment with DA for 36 h in HCT-116^MDR^ cells; (**C**) ATPase activity was measured in HCT-116^MDR^ after treatment with DA for 36 h; (**D**) expression of intracellular fumarate levels in HCT-116^MDR^ after treatment with DA for 36 h; (**E**) expression of α-ketoglutarate levels in HCT-116^MDR^ after treatment with DA for 36 h. Densitometry analysis for all the proteins was normalized with β-actin and measured using Image J software. The representative data shown here are the means ± S.D. from three independent experiments where * *p* < 0.05, ** *p* < 0.01 and *** *p* < 0.001. HCT-116^MDR^ vs. all other groups calculated through ANOVA prism. Original western blots are presented in Appendix A.

**Figure 5 cancers-15-03541-f005:**
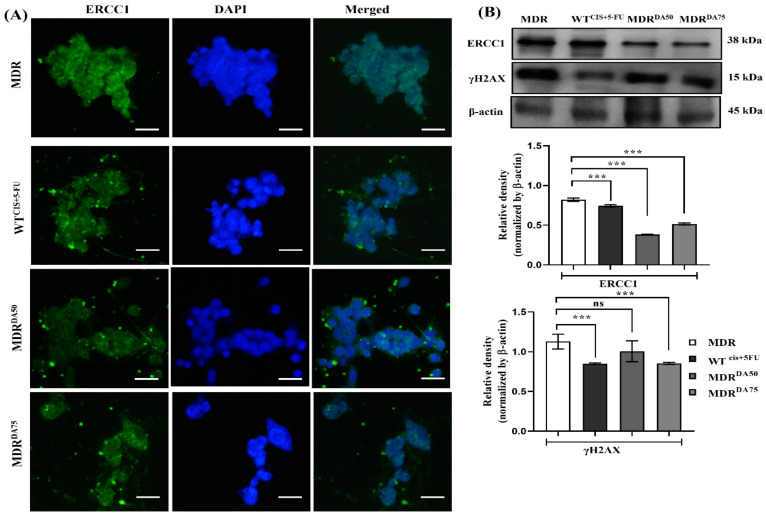
DA treatment downregulated the expression of DNA damage repair in HCT-116^MDR^ CRC cells. (**A**) Immunofluorescence staining of ERCC1 in HCT-116^MDR^ cells after treatment with DA for 36 h (magnification: 600×, scale bar: 25 μm); (**B**) Western blotting analysis of DNA damage repair markers after treatment with DA for 36 h in HCT-116^MDR^ cells. Densitometry analysis for all the proteins was normalized with β-actin and measured using Image J software. The representative data shown here are the means ± S.D. from three independent experiments where *** *p* < 0.001. HCT-116^MDR^ vs. all other groups calculated through ANOVA prism. Original western blots are presented in Appendix A.

**Figure 6 cancers-15-03541-f006:**
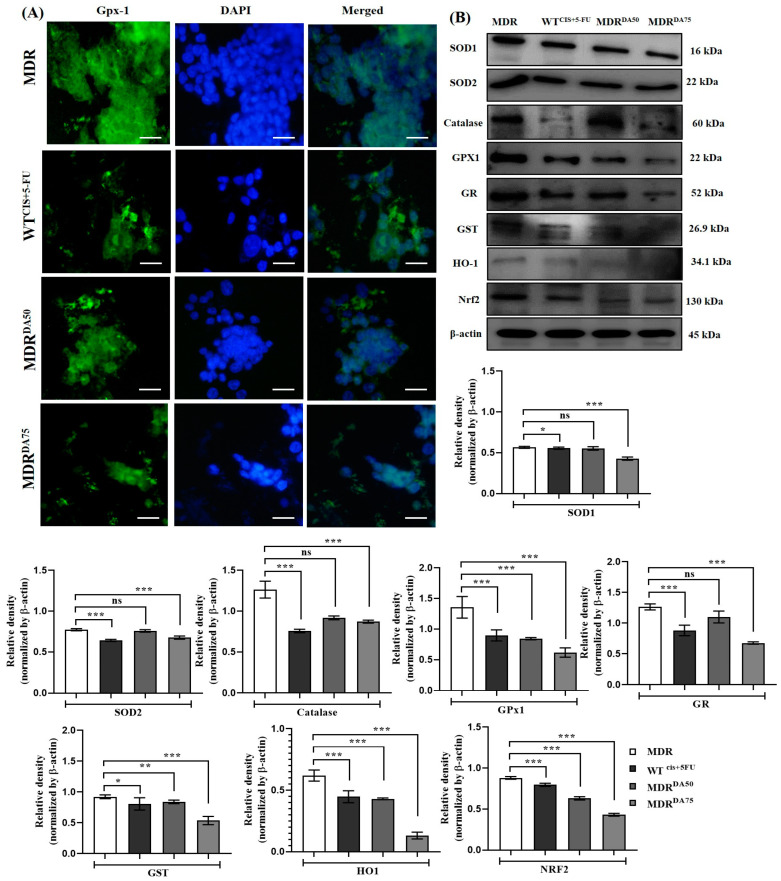
DA treatment significantly downregulated the expression of antioxidants in HCT-116^MDR^ CRC cells. (**A**) Immunofluorescence staining of Gpx-1 in HCT-116^MDR^ cells after treatment with DA for 36 h (magnification: 600×, scale bar: 25 μm); (**B**) Western blotting analysis of antioxidant markers after treatment with DA for 36 h in HCT-116^MDR^ cells. Densitometry analysis for all the proteins was normalized with β-actin and measured using Image J software. The representative data shown here are the means ± S.D. from three independent experiments where * *p* < 0.05, ** *p* < 0.01 and *** *p* < 0.001. HCT-116^MDR^ vs. all other groups calculated through ANOVA prism. Original western blots are presented in Appendix A.

**Figure 7 cancers-15-03541-f007:**
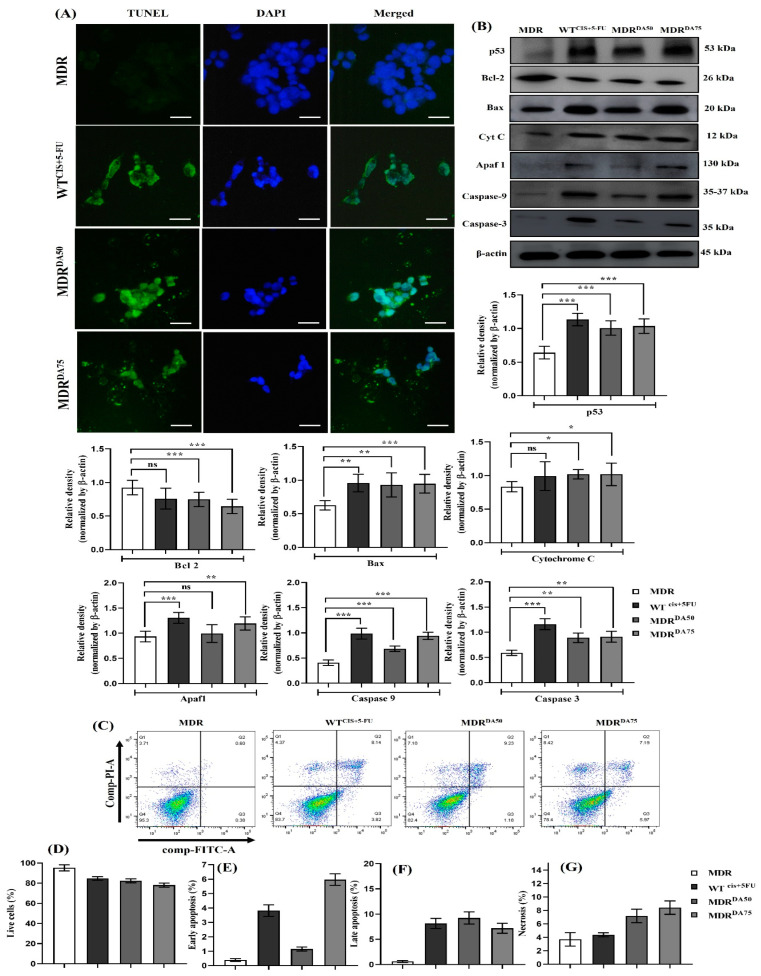
DA treatment triggered intrinsic apoptotic cell death in HCT-116^MDR^ CRC cells. (**A**) TUNEL assay staining in HCT-116^MDR^ cells after treatment with DA for 36 h (magnification: 600×, scale bar: 25 μm); (**B**) Western blotting analysis of intrinsic apoptosis markers after treatment with DA for 36 h in HCT-116^MDR^ cells; (**C**) annexin V/propidium iodide flow cytometry evaluation of apoptosis after treatment with different concentrations of DA for 36 h; (**D**) percentage of live cells after DA treatment evaluated through flow cytometry; (**E**) percentage of early apoptosis cells after DA treatment evaluated through flow cytometry; (**F**) percentage of late apoptosis cells after DA treatment evaluated through flow cytometry; (**G**) percentage of necrotic cells after DA treatment evaluated through flow cytometry. Densitometry analysis for all the proteins was normalized with β-actin and measured using Image J software. The representative data shown here are the means ± S.D. from three independent experiments where * *p* < 0.05, ** *p* < 0.01 and *** *p* < 0.001. HCT-116^MDR^ vs. all other groups calculated through ANOVA prism. Original western blots are presented in Appendix A.

**Figure 8 cancers-15-03541-f008:**
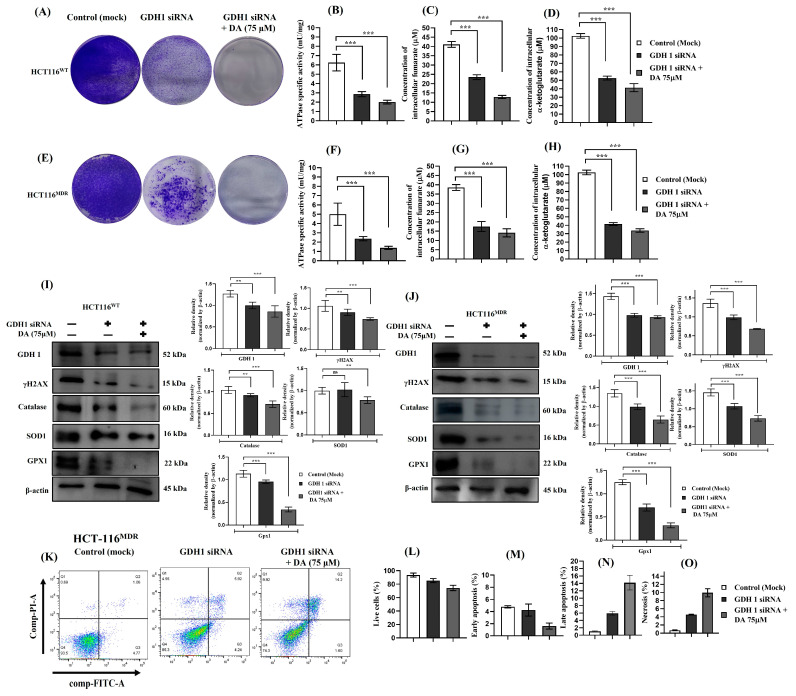
Knockdown of glutamate dehydrogenase 1 in HCT-116^MDR^ CRC cells resulted in reduced cell proliferation and antioxidant depletion. (**A**) Evaluation of colony-forming rate and cell proliferation in parental HCT-116^WT^ cells after GDH1 silencing and treatment with DA for 36 h; (**B**) ATPase activity was measured in HCT-116^WT^ after GDH1 silencing and treatment with DA for 36 h; (**C**) expression of intracellular fumarate levels was measured in HCT-116^WT^ after GDH1 silencing and treatment with DA for 36 h; (**D**) expression of α-ketoglutarate levels was measured in HCT-116^WT^ after GDH1 silencing and treatment with DA for 36 h; (**E**) evaluation of colony-forming rate and cell proliferation in HCT-116^MDR^ cells after GDH1 silencing and treatment with DA for 36 h; (**F**) ATPase activity was measured in HCT-116^MDR^ cells after GDH1 silencing and treatment with DA for 36 h; (**G**) expression of intracellular fumarate levels was measured in HCT-116^MDR^ cells after GDH1 silencing and treatment with DA for 36 h; (**H**) expression of α-ketoglutarate levels was measured in HCT-116^MDR^ cells after GDH1 silencing and treatment with DA for 36 h; (**I**) Western blotting analysis of DNA damage repair and antioxidant markers after GDH1 silencing and treatment with DA for 36 h in HCT-116^WT^ cells; (**J**) Western blotting analysis of DNA damage repair and antioxidant markers after GDH1 silencing and treatment with DA for 36 h in HCT-116^MDR^ cells; (**K**) annexin V/propidium iodide flow cytometry evaluation of apoptosis after GDH1 silencing and treatment with DA for 36 h in HCT-116^MDR^ cells; (**L**) percentage of live cells after GDH1 silencing in HCT-116^MDR^ cells evaluated through flow cytometry; (**M**) percentage of early apoptosis cells after GDH1 silencing in HCT-116^MDR^ cells evaluated through flow cytometry; (**N**) percentage of late apoptosis cells after GDH1 silencing in HCT-116^MDR^ cells evaluated through flow cytometry; (**O**) percentage of necrotic cells after GDH1 silencing in HCT-116^MDR^ cells evaluated through flow cytometry. Densitometry analysis for all the proteins was normalized with β-actin and measured using Image J software. The representative data shown here are the means ± S.D. from three independent experiments where ** *p* < 0.01 and *** *p* < 0.001. HCT-116^MDR^ (mock) vs. all other groups calculated through ANOVA prism. Original western blots are presented in Appendix A.

## Data Availability

The datas presented in this study will be available upon reasonable request from the corresponding author.

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
