# Peer review of "Decursinol Angelate Inhibits Glutamate Dehydrogenase 1 Activity and Induces Intrinsic Apoptosis in MDR-CRC Cells"

_cancers, 2023, doi:10.3390/cancers15143541_

Round 1
Reviewer 1 Report
None
Author Response
- Author response to the comments of Reviewer 1
Comments and Suggestions for Authors: None
Author Response: Thank you for considering our manuscript to be published in this esteemed cancers journals. We the authors highly regard your opinions and reviewing our manuscript and our sincerest gratitude for you.

Reviewer 2 Report
We consider the manuscript pertinent to the readers of this Journal.
This study is a continuation of the team and focuses on the role of decursinol angelate (DA) in colorectal cancer (CRC). The novelty of the manuscript is the clarification of DA's inhibitory role towards glutamate dehydrogenase and its apoptosis effect in multidrug-resistant CRC cell lines.
The introduction briefly covers old and new references and perfectly integrates the theme's main aspects.
This article has a good organization of the contents. Regarding the discussion of the results, we found it suitable. A very nice graphic pics/graphs quality accompanying the discussion increases the understanding of the discussed theme and clarifies the reasoning. We congratulate the authors on that!
The core experimental design of the chemistry part of the manuscript seems carefully elaborated and meticulously presented.
The statistical analysis aiming for the comparison of the mean differences for experimental groups of WT Cis+5fu and HCT-116MDR, MDRDA50, MDRDA75, NAC groups, and siRNA groups was appropriately applied. However, regarding the methodology of inhibition assays, some doubts were raised.
The description of the enzyme kinetic studies is insufficient to guarantee reproducibility and some conclusions (e.g. IC50) presented in the abstract don’t have support in the results and discussion.
To be noticed, it is necessary to make a brief description of some fundamental enzymological points, such as substrate concentrations, inhibitor concentrations, and enzyme concentrations. It should be clarified also how the enzyme activity was defined in the assay.
Specific Comments:
#1__L19-L20_GDH acronym must be defined, as well as all the others throughout the manuscript. Dear authors, please review for the correct definition of acronyms the first time they appear.
#2_ Enzyme Commission number (EC number) of the enzymes used. The Enzyme Commission number (EC number) is a system of enzyme nomenclature, every EC number is associated with a recommended name for the corresponding enzyme-catalyzed reaction to be unequivocally identified. Please, dear authors, include de EC number to identify clearly the enzyme used and the reaction catalyzed.
#3_ L25 and L26__” DA treatment significantly downregulated the expression of GDH1 at 50 and 75 μM after 36 h, which directly correlated with reduced.” The term “significantly” or “significant” generally is used whenever an appropriate statistical test was applied and there is a level of significance (p-value). what do the values 50-57 MU refer to? To the IC50? Please check for this.
#4 _ L471 and L573_” from the obtained results….” and “DA showed inhibition against glutamate...). Please clarify the support of the reasoning that there was effective inhibition of the enzyme or it´s only hypothesized by the authors (?). The inhibition conclusion of the enzyme ought to be supported by inhibition parameters (IC50, Ki,..) and we didn’t find any.
Author Response
- Author response to the comments of Reviewer 2
Comments and Suggestions for Authors
General Remark: We consider the manuscript pertinent to the readers of this Journal.
This study is a continuation of the team and focuses on the role of decursinol angelate (DA) in colorectal cancer (CRC). The novelty of the manuscript is the clarification of DA's inhibitory role towards glutamate dehydrogenase and its apoptosis effect in multidrug-resistant CRC cell lines.
The introduction briefly covers old and new references and perfectly integrates the theme's main aspects.
This article has a good organization of the contents. Regarding the discussion of the results, we found it suitable. A very nice graphic pics/graphs quality accompanying the discussion increases the understanding of the discussed theme and clarifies the reasoning. We congratulate the authors on that!
The core experimental design of the chemistry part of the manuscript seems carefully elaborated and meticulously presented.
The statistical analysis aiming for the comparison of the mean differences for experimental groups of WT Cis+5fu and HCT-116MDR, MDRDA50, MDRDA75, NAC groups, and siRNA groups was appropriately applied. However, regarding the methodology of inhibition assays, some doubts were raised.
Author Response: Thank you for your support for our submitted manuscript. We hold your review comments on high regards and our sincerest gratitude for you.
Query 1: The description of the enzyme kinetic studies is insufficient to guarantee reproducibility and some conclusions (e.g. IC50) presented in the abstract don’t have support in the results and discussion.
Author Response: Thank you for your insightful comments, Sir. We have previously published our in silico and enzymatic assay findings for the role of decursinol angelate (DA) to inhibit the activity of human GDH by binding to the abortive complex of the ADP activation site. In our previous article, we performed molecular docking analysis of 8 different single compounds collected from pubChem database isolated from different plant sources and selected decursin (DN) and decursinol angelate (DA). We performed molecular dynamics simulation (MD), monitored the stability, interaction for protein and docked ligand at 50 ns, and evaluated the molecular mechanics Poisson-Boltzmann surface area (MM-PBSA) free energy calculation on the two chosen compounds along with a standard inhibitor epigallocatechin gallate (EGCG) as reference. The final results showed the formation of stable hydrogen bond interactions by DN and DA in the residues of R400 (arginine 400) and Y386 (tyrosine 386) at the ADP activation site of GDH, which was important for the selective inhibition of GDH activity. Additionally, the total binding energy of DN (decursin) and DA were -115.5 KJ/mol and -106.2 KJ/mol, which was higher than the standard reference GDH inhibitor EGCG (-92.8 KJ/mol). Furthermore, biochemical analysis for GDH inhibition substantiated our computational results and established DN and DA as novel GDH inhibitor. The percentage of IC50 inhibition for DN and DA were 1.035 μM and 1.432 μM. Conclusively, DN and DA can be a novel therapeutic drug for inhibition of glutamate dehydrogenase. These findings have not been included in our current study of DA on HCT-116 MDR cancer cells.
Query 2: To be noticed, it is necessary to make a brief description of some fundamental enzymological points, such as substrate concentrations, inhibitor concentrations, and enzyme concentrations. It should be clarified also how the enzyme activity was defined in the assay.
Author Response: Thank you for your insightful comments, Sir. In our previously published paper, we have performed the enzymatic assay which is described as such.
Human GDH activity assay
For performing the activity of GDH, we first optimized the concentration by measuring the linear dose response of the enzymatic reaction initial velocities. We used different dilutions of GDH (50, 100, 200, 300, 400, 500, 1000 nM) and they were added to the biochemical 96 well assay plates along with the GDH reaction buffer which was prepared as per the following concentration (50 mM of Tris-HCl, pH 8, 0.01% bovine serum albumin, 0.001% tween 20, 0.003% Brij-35, pH 8). Next, we added glutamate (5 mM), NADP+ (100 μM) and EZMTT detection reagent (0.5 mM EZMTT in 100 μL of Tris-HCl). After addition of all the chemicals, the 96 well plates were incubated at room temperature for the necessary chemical reaction to be carried out and the absorbance was measured after 2h at 450 nm. For a time-dependent and dose-dependent study absorbance was measured every 10 min for 1 hour to collect linear initial velocity at 450 nm. For the GDH inhibition assay in 50 mM Tris-HCl (pH 8.2) at 25︒C, 3-fold dilutions of compounds (0 - 26 μM) were incubated with the GDH enzyme for 0.5 hours before the addition of the precursors to start the reaction. After 0.5h, a mixture of glutamate (5 mM final), NADP+ (200 μM final) and the EZMTT detection reagent were added and reacted for 2 h to measure the GDH activity.
Compound GDH inhibition assay
For measuring the inhibition of human GDH, we used a series of different dilutions for the inhibitors EGCG and DA (1-10 μM) was added onto the 96 well biochemical assay plates and were mixed with the human GDH (10 nM) with the GDH buffer (50 mM of Tris-HCl, pH 8, 0.01% bovine serum albumin, 0.001% tween 20, 0.003% Brij-35, pH 8). After 0.5 hour of incubation with the compound and the GDH, to each wells we added glutamate (5 mM), NADP+ (100 μM final) and EZMTT detection reagent. After the addition of the chemicals, the 96 well plates were incubated for 2h in room temperature. After incubation period was completed, the absorbance was measured at 450 nm (reference wavelength 620 nm). Additionally, we measured the activity of the compounds EGCG and DA by incubating it with or without the coenzyme NAD+ or NADP+. Even for the activity of GDH to be used as a control. We measured the activity of GDH with or without the NAD+ or NADP+.
The detailed description and results are published in our manuscript mentioned below:
Reference:
“Chang, S.N.; Keretsu, S.; Kang, S.C. Evaluation of Decursin and Its Isomer Decursinol Angelate as Potential Inhibitors of Human Glutamate Dehydrogenase Activity through in Silico and Enzymatic Assay Screening. Comput. Biol. Med. 2022, doi:10.1016/j.compbiomed.2022.106287”.
Specific Comments:
Query 3: #1__L19-L20_GDH acronym must be defined, as well as all the others throughout the manuscript. Dear authors, please review for the correct definition of acronyms the first time they appear.
Author Response: Thank you for your insightful comments, Sir. We have provided the details for all the acronyms when it was described the first time in the manuscript. We did not perform any enzymatic assay for this study and have just used glutamate dehydrogenase 1 (GDH1) antibody which was purchased from cell signalling for evaluating the immunofluorescence staining and western blotting analysis.
Query 4: #2_ Enzyme Commission number (EC number) of the enzymes used. The Enzyme Commission number (EC number) is a system of enzyme nomenclature, every EC number is associated with a recommended name for the corresponding enzyme-catalyzed reaction to be unequivocally identified. Please, dear authors, include de EC number to identify clearly the enzyme used and the reaction catalyzed.
Author Response: Thank you for your insightful comments, Sir. The enzyme commission number for glutamate dehydrogenase is EC 1.4. 1.2.
Query 5: #3_ L25 and L26__” DA treatment significantly downregulated the expression of GDH1 at 50 and 75 μM after 36 h, which directly correlated with reduced.” The term “significantly” or “significant” generally is used whenever an appropriate statistical test was applied and there is a level of significance (p-value). what do the values 50-57 MU refer to? To the IC50? Please check for this.
Author Response: Thank you for your insightful comments, Sir. We have added significant to the sentence while describing the results as there was a statistical significance found from the densitometric analysis of western blotting markers for the markers. 50-75 MU is 50 µM and 75 µM of decursinol angelate (DA). After evaluating the cell viability assay at different time points after treatment with DA, we observed that the IC50 of DA was 75 µM with 50 percent of cell dying. So, we used 50 µM and 7550 µM dose to carry out our experimental study for the HCT-116 MDR manuscript.
Query 6: #4 _ L471 and L573_” from the obtained results….” and “DA showed inhibition against glutamate...). Please clarify the support of the reasoning that there was effective inhibition of the enzyme or it´s only hypothesized by the authors (?). The inhibition conclusion of the enzyme ought to be supported by inhibition parameters (IC50, Ki,..) and we didn’t find any.
Author Response: Thank you for your insightful comments, Sir. The inhibition of glutamate dehydrogenase by decursinol angelate (DA) was proven from our previously published manuscript which I have cited in query 2 as well. The IC50 for the inhibition of GDH by DA was 1.432 μM and the total binding energy of DA with GDH was -106.2 KJ/mol. DA also formed multiple interactions in the GDH ADP activation site

Reviewer 3 Report
The title of the manuscript is good. English language has good quality. Figures have acceptable quality. Some sections of the manuscript need some changes.
1. About the part simple summary:
Please reconsider this part because:
+ this part is a little complex and hard to understand
+ this part should contain a brief and simple introduction about present study but it has some information about the previous work of authors
+ this part has some abbreviations that need to be expanded and defined
2. Please reconsider the part "Abstract" based on order below:
+ First: write briefly about the importance of colorectal cancer
+ second: write briefly about the importance of treatment of colorectal cancer
+ write briefly about the importance of your present work
+ write briefly about material and method
+ write briefly about results
+ write briefly about conclusion
3. Line 39-40 in page 1
Please add proper reference at the end of this sentence
4. Line 46-47 in page 2
The authors have mentioned "Another research study conducted in 2020 showed around 147,950 new cases of CRC di-46 agnosed in people and 53,200 individuals died from the disease."
What do you mean by "Another research study"? Is there any survey before this study? If yes, please mention it before this sentence.
5. Line 52 in page 2 the authors have suddenly spoken about the treatment of CRC
but in the previous sentence, they have talked about risk factors of CRC. In better words, line 52 has no connection with its prior sentence. Please create logical link between this line and its prior sentence.
6. In the section "Introduction" there are some multipple references. Please reform them.
7. Line 67 in page 2 has no proper link with its previous paragraph (sentence). Please make suitable link.
8. Line 109 in page 3
Why this line has two references?
9. It is better to insert the amount of scale bar in each figure in addition to insert it in the title of each figure.
10. Line 312 - 315 belongs to other parts of the manuscript. Please remove it from the
section "Results".
11. Line 323-326 should be removed from this part of manuscript
Please note that in the secrion "Results", you should only talk about results. Any other information that belongs to other parts of manuscript like "Discussion" or "Material and methods" and also any sentence with reference should be removed from this part.
12. All over the section "Results" there are some sentence with citation. Please remove all of them and refer to comment 11.
13. About the part "Discussion"
Please rewrite this part according to notes below:
First: categorize all of your results based on their importance (from the most important one to the least important)
Second: after that, turn each one of your results into some subheadings
Third: after that, discuss about them one by one
Forth: make comparisons between your results and the results of other similar and relevant surveys
14. This is important that the authors compare their findings with the results of prior similar surveys in the part "Discussion".
15. Please check and adjust the "Reference list" based on the regulations of reference list of journal. (Titles, doi, the name of journal and ... )
Author Response
- Author response to the comments of Reviewer 3
General Remark: The title of the manuscript is good. English language has good quality. Figures have acceptable quality. Some sections of the manuscript need some changes.
Author Response: Thank you for considering our manuscript to be published in this esteemed cancers journals. We the authors highly regard your opinions and reviewing our manuscript and our sincerest gratitude for you.
Query: 1. About the part simple summary:
Please reconsider this part because:
+ this part is a little complex and hard to understand
+ this part should contain a brief and simple introduction about present study but it has some information about the previous work of authors
+ this part has some abbreviations that need to be expanded and defined
Author Response: Thank you for your insightful comments. We have added a brief description of the present study in the brief summary and has also expanded the abbreviations mentioned in the short summary. The changes can be found in Line no: 8, 9, 17-18.
Query: 2. Please reconsider the part "Abstract" based on order below:
+ First: write briefly about the importance of colorectal cancer
+ second: write briefly about the importance of treatment of colorectal cancer
+ write briefly about the importance of your present work
+ write briefly about material and method
+ write briefly about results
+ write briefly about conclusion
Author Response: Thank you for your insightful comments. We have made minor changes to the abstract as per your request and the changes can be found in Line number 20-26, 29, 31, 33, 37.
Query: 3. Line 39-40 in page 1
Please add proper reference at the end of this sentence
Author Response: Thank you for your insightful comments. The reference for the paper 39-40 is the same as the reference mentioned in line 42 reference 1.
Query: 4. Line 46-47 in page 2
The authors have mentioned "Another research study conducted in 2020 showed around 147,950 new cases of CRC diagnosed in people and 53,200 individuals died from the disease."
What do you mean by "Another research study"? Is there any survey before this study? If yes, please mention it before this sentence.
Author Response: Thank you for your insightful comments. We have changed the sentence and updated it accordingly as per the research paper from where we referenced it. The sentence has been edited this way in our revised manuscript “According to the National Center for health statistics updated by the American Cancer Society, around 147,950 new cases of CRC diagnosed in people and 53,200 individuals died from the disease in 2020”. The new changes can be found in the manuscript Line number: 54-55.
Query: 5. Line 52 in page 2 the authors have suddenly spoken about the treatment of CRC
but in the previous sentence, they have talked about risk factors of CRC. In better words, line 52 has no connection with its prior sentence. Please create logical link between this line and its prior sentence.
Author Response: Thank you for your insightful comments. Line 52 which in the revised manuscript is Line number 61 is a new paragraph where we have started a new topic for the treatment of colorectal cancer. In the previous paragraph like you have mentioned, we wrote about the occurrence of CRC and the statistics provided by the National Center for health statistics updated by the American Cancer Society.
Query: 6. In the section "Introduction" there are some multipple references. Please reform them.
Author Response: Thank you for your comments. These references were important for structuring our manuscript with the fundamentals which were acquired from the previous research studies conducted by different authors renowned in the field of CRC and other cancer research scientists. I have made minor modifications to the introduction section but I have retained the references as they form the integral part from where I collected information to write the manuscript. I have also added additional information to the manuscript which can be found in the revised manuscript Line number 60-67. The new addition is detailed here “As per the data collected from the clinical proteomic tumor analysis consortium (CPTAC) (http://ualcan.path.uab.edu) for GDH expression in CRC patients with respect to different parameters such as different stages of cancer, stage 3 CRC patients were found to have higher GDH expression. Similarly, GDH expression in CRC patients according to the different races revealed African-Americans CRC patients to have higher GDH expression than the other races of CRC patients from Asian and Caucasian origin. African-American CRC patients were also found to be highly susceptible with high mortality rate”.
Query: 7. Line 67 in page 2 has no proper link with its previous paragraph (sentence). Please make suitable link.
Author Response: Thank you for your useful comments. We have changed the sentence to make a suitable link with the continuation from the previous paragraph. The changes made are mentioned here “Altered cellular metabolism favouring proliferation, growth, maintenance, and survival, irrespective of the type of cells or tissues is observed across different cancers”. The changes can be found in the revised manuscript in Line number 83-84.
Query: 8. Line 109 in page 3
Why this line has two references?
Author Response: Thank you for your useful comments. These two references are the published papers from the laboratory where I conducted the research work. These references were used for generating a drug resistant HCT-116 cancer cell line for studies conducted in this manuscript.
Query: 9. It is better to insert the amount of scale bar in each figure in addition to insert it in the title of each figure.
Author Response: Thank you for your insightful comments. I do not have the necessary tools and software right now to edit the scale bars outside the figures. It is my sincere request to please accept the figures scale bar as it is.
Query: 10. Line 312 - 315 belongs to other parts of the manuscript. Please remove it from the section "Results".
Author Response: Thank you for your insightful comments. As per your suggestion the sentences from line 312-315 has been removed from the result section. The removed sections in the revised manuscript is in the Line number: 328.
Query: 11. Line 323-326 should be removed from this part of manuscript
Please note that in the secrion "Results", you should only talk about results. Any other information that belongs to other parts of manuscript like "Discussion" or "Material and methods" and also any sentence with reference should be removed from this part.
Author Response: Thank you for your insightful comments. The Line no 323-326 has been removed from result section and has been moved to discussion section. The changes can be found in the revised manuscript Line number: 523-530.
Query: 12. All over the section "Results" there are some sentence with citation. Please remove all of them and refer to comment 11.
Author Response: Thank you for your insightful comments. All the result sections that we had added references were removed and moved to discussion section. The changes can be found in discussion in the revised manuscript Line number: 487-488, 490-493, 505-511, 520-522, 523-539.
Query: 13. About the part "Discussion"
Please rewrite this part according to notes below:
First: categorize all of your results based on their importance (from the most important one to the least important)
Second: after that, turn each one of your results into some subheadings
Third: after that, discuss about them one by one
Forth: make comparisons between your results and the results of other similar and relevant surveys
Author Response: Thank you for your insightful comments. I have made the necessary changes as per your suggestion. The changes can be found in discussion in the revised manuscript Line number: 487-488, 490-493, 505-511, 520-522, 523-539.
Query: 14. This is important that the authors compare their findings with the results of prior similar surveys in the part "Discussion".
Author Response: Thank you for your insightful comments. We have compared our results with the studies from previous researchers. The comparison for the references can be found in the revised manuscript from reference number 32-55.
Query: 15. Please check and adjust the "Reference list" based on the regulations of reference list of journal. (Titles, doi, the name of journal and ... )
Author Response: Thank you for your useful comments. We have adjusted the references according to the cancers (MDPI) format from Mendley software for citation.

Reviewer 4 Report
Overview and general recommendation:
In this study, the authors develop a multidrug-resistant cell line for cisplatin and 5-fluorouracil (HCT-116MDR) with HCT-116WT. the authors investigate the phenotypes and molecular characteristics of HCT-116MDR by showing the morphology, extracellular vesicles, testing expression of markers of drug resistance, cell proliferation, DNA damage response, and GDH1, an enzyme involved in glutamine metabolism. Then they show that with decursinol angelate (DA) treatment, the expression of multidrug-resistant markers and DNA damage repair markers and GDH1 are down-regulated in HCT-116MDR. They also show that DA enhance apoptosis in HCT-116MDR CRC cells.
I find the paper is organized in a proper way and the results are well described. The authors performed detailed background research and the research is designed rationally. And major methods are well described and properly used in the research. The result is good enough to support their conclusions. Figures are organized in a proper way.
Major comments:
1. In result 3.1, the authors claim that “ERCC1 expression (p< ns) was upregulated in both HCT-116 WT and HCT-116 MDR cells”. How do you know the expression was upgulated in both cell lines? Do you have a control showing the upgulation?
2. In Fig 3A, image of wt without cis and 5-fu treatment should be present here as a control.
Minor comments:
1. Page1 line40, I think it should be “….aggressively spread to….”
2. Page2 line56, I think it should be”…. proven to be effective….”
3. Page2 line57, I think it should be”…. have shown significant efficacy….”
4. Page2 line59, I think it should be”…. which remain major challenges….”
5. Page2 line86, I think it should be”…. in melanoma in vitro and in vivo….”
6. Page18 line535-538, I think the font should be fixed.
good
Author Response
- Author response to the comments of Reviewer 4
Comments and Suggestions for Authors
General Remark: In this study, the authors develop a multidrug-resistant cell line for cisplatin and 5-fluorouracil (HCT-116MDR) with HCT-116WT. the authors investigate the phenotypes and molecular characteristics of HCT-116MDR by showing the morphology, extracellular vesicles, testing expression of markers of drug resistance, cell proliferation, DNA damage response, and GDH1, an enzyme involved in glutamine metabolism. Then they show that with decursinol angelate (DA) treatment, the expression of multidrug-resistant markers and DNA damage repair markers and GDH1 are down-regulated in HCT-116MDR. They also show that DA enhances apoptosis in HCT-116MDR CRC cells.
I find the paper is organized in a proper way and the results are well described. The authors performed detailed background research and the research is designed rationally. And major methods are well described and properly used in the research. The result is good enough to support their conclusions. Figures are organized in a proper way.
Author Response: Thank you for your insightful comments. We the authors highly regard your opinions and reviewing our manuscript.
Major comments:
Query: 1. In result 3.1, the authors claim that “ERCC1 expression (p< ns) was upregulated in both HCT-116 WT and HCT-116 MDR cells”. How do you know the expression was upregulated in both cell lines? Do you have a control showing the upregulation?
Author Response: Thank you for your insightful comments. We did not evaluate the expression on ERCC1 on normal colon cell line and just compared the HCT-116 wild type and HCT-116 multi-drug resistant cell line for this study and we apologize to the reviewers for this. As per the studies from other researchers, the excision repair cross-complementation group 1 (ERCC1) enzyme plays a rate-limiting role in the nucleotide excision repair pathway that recognizes and removes cisplatin-induced DNA adducts [1]. ERCC1 is also important in the repair of interstrand cross-links in DNA and in recombination processes [2]. In vitro studies have linked platinum resistance to the expression of ERCC1 messenger RNA (mRNA) in cell lines involved in ovarian, cervical, testicular, bladder, and non–small-cell lung cancers [3]. The relation between the expression of ERCC1 mRNA and resistance to platinum compounds has been corroborated by small, retrospective clinical studies in patients with advanced gastric, ovarian, colorectal, esophageal, or non–small-cell lung cancer [4].
Reference
- Sancar, A. Mechanisms of DNA Excision Repair. Science (80-. ).
- Laura J. Niedernhofer, Hanny Odijk, Magda Budzowska, Ellen van Drunen, Alex Maas, Arjan F. Theil, Jan de Wit, N. G. J. Jaspers, H. Berna Beverloo, Jan H. J. Hoeijmakers & Roland Kanaar(2004) The Structure-Specific Endonuclease Ercc1-Xpf Is Required To Resolve DNA Interstrand Cross-Link-Induced Double-Strand Breaks, Molecular and Cellular Biology, 24:13, 5776-5787, DOI: 1128/MCB.24.13.5776-5787.2004
- Altaha R, Liang X, Yu JJ, Reed E. Excision repair cross complementing-group 1: gene expression and platinum resistance. Int J Mol Med2004;14:959-970
- Metzger R, Leichman CG, Danenberg KD, et al. ERCC1 mRNA levels complement thymidylate synthase mRNA levels in predicting response and survival for gastric cancer patients receiving combination cisplatin and fluorouracil chemotherapy. J Clin Oncol1998;16:309-316
Query: 2. In Fig 3A, image of wt without cis and 5-fu treatment should be present here as a control.
Author Response: Thank you for your insightful comments. We have made the changes as per your suggestion. The new changes in the grouping on WTcis +5Fu is now HCT-116 WTCis+5Fu in Fig. 3A.
Minor comments:
Query 1. Page1 line40, I think it should be “….aggressively spread to….”
Author Response: Thank you for your comments. The changes has been made as per suggested and it can be found in revised manuscript Line number: 48
Query 2. Page2 line56, I think it should be”…. proven to be effective….”
Author Response: Thank you for your comments. The changes has been made as per suggested and it can be found in revised manuscript Line number: 72
Query 3. Page2 line57, I think it should be”…. have shown significant efficacy….”
Author Response: Thank you for your insightful comments. The changes has been made as per suggested and it can be found in revised manuscript Line number: 73
Query 4. Page2 line59, I think it should be”…. which remain major challenges….”
Author Response: Thank you for your insightful comments. The changes has been made as per suggested and it can be found in revised manuscript Line number: 75
Query 5. Page2 line86, I think it should be”…. in melanoma in vitro and in vivo….”
Author Response: Thank you for your insightful comments. The changes has been made as per suggested and it can be found in revised manuscript Line number: 102
Query 6. Page18 line535-538, I think the font should be fixed.
Author Response: Thank you for your noticing the mistake. We have corrected the the font size according to the journal format. The changes in the revised manuscript can be found in Line number: 549-553.

Round 2
Reviewer 2 Report
We appreciate the efforts of the authors to respond in detail to our questions.
Dear authors, it seems to us that some changes still do not appear in this latest version of the manuscript, e.g. inclusion of the EC Number of the enzyme. Please check if all the changes reviewed appear in the latest version.
We just have to remember that the results of previous publications must be referred to in the discussion with the proper citation of the work, to prevent the reader from considering these results as obtained in the present work.
We await more detailed studies for this enzyme in your future work, regarding its real mechanism. The IC50 is completely devoid of information about the mechanism, we advise estimating the inhibition kinetic constants to better understand the mechanism involved!
Good job!
Author Response
- Author response to the 2nd revised comments of Reviewer 2
Comments and Suggestions for Authors
General Remark: We appreciate the efforts of the authors to respond in detail to our questions.
Dear authors, it seems to us that some changes still do not appear in this latest version of the manuscript, e.g. inclusion of the EC Number of the enzyme. Please check if all the changes reviewed appear in the latest version.
Author Response: Thank you for your important suggestions for improving our submitted manuscript. We have included the EC number for glutamate dehydrogenase in our second revised manuscript which can be found in Line number: 115-116.
We just have to remember that the results of previous publications must be referred to in the discussion with the proper citation of the work, to prevent the reader from considering these results as obtained in the present work.
Author Response: Thank you for your support for our submitted manuscript. In order to avoid the confusion to the revised manuscript, we have mentioned the results of our study previous publication in simple summary (Line number: 8-9), Abstract section (Line number: 23-24), introduction (Line number: 60-67, 102-103). The reference number 20 in the manuscript of my previous publication has been cited 3 times in the manuscript and the citations can be found in Line number: 67, 103,544.
We await more detailed studies for this enzyme in your future work, regarding its real mechanism. The IC50 is completely devoid of information about the mechanism, we advise estimating the inhibition kinetic constants to better understand the mechanism involved!
Good job!
Author Response: Thank you for your support for our submitted manuscript. We will perform more studies in the future on the molecular mechanism mode of study on this question raised. We will also estimate the inhibition kinetic constant in our next study. We hold your review comments on high regards and our sincerest gratitude for you.

Reviewer 3 Report
There is no more comments.
Author Response
Thank you for considering our manuscript to be published in the reputed cancers journal. We thank you the reviewer for reviewing our manuscript and suggesting important changes to the manuscript with your expertise.